# Discovering SNP-disease relationships in genome-wide SNP data using an improved harmony search based on SNP locus and genetic inheritance patterns

**Fariba Esmaeili, Zahra Narimani** *, **Mahdi Vasighi**

Department of Computer Science and Information Technology, Institute for Advanced Studies in Basic Sciences (IASBS), Zanjan, Iran

* narimani@iasbs.ac.ir

**Data Availability Statement:** Codes are available via: https://github.com/BioLab-IASBS/LDHS Whole dataset is available via figshare (DOI: 10.6084/m9. figshare.24038325).

## Abstract

Advances in high-throughput sequencing technologies have made it possible to access millions of measurements from thousands of people. Single nucleotide polymorphisms (SNPs), the most common type of mutation in the human genome, have been shown to play a significant role in the development of complex and multifactorial diseases. However, studying the synergistic interactions between different SNPs in explaining multifactorial diseases is challenging due to the high dimensionality of the data and methodological complexities. Existing solutions often use a multi-objective approach based on metaheuristic optimization algorithms such as harmony search. However, previous studies have shown that using a multi-objective approach is not sufficient to address complex disease models with no or low marginal effect. In this research, we introduce a locus-driven harmony search (LDHS), an improved harmony search algorithm that focuses on using SNP locus information and genetic inheritance patterns to initialize harmony memories. The proposed method integrates biological knowledge to improve harmony memory initialization by adding SNP combinations that are likely candidates for interaction and disease causation. Using a SNP grouping process, LDHS generates harmonies that include SNPs with a higher potential for interaction, resulting in greater power in detecting disease-causing SNP combinations. The performance of the proposed algorithm was evaluated on 200 synthesized datasets for disease models with and without marginal effect. The results show significant improvement in the power of the algorithm to find disease-related SNP sets while decreasing computational cost compared to state-of-the-art algorithms. The proposed algorithm also demonstrated notable performance on real breast cancer data, showing that integrating prior knowledge can significantly improve the process of detecting disease-related SNPs in both real and synthesized data.

**Funding:** The author(s) received no specific funding for this work.

**Competing interests:** The authors have declared that no competing interests exist.

## 1 Introduction

SNPs are the most common cause of genetic diversity in humans and can be identified as single nucleotide differences in the same location in the human genome. On average, there is one SNP per thousand nucleotides in the human genome, and each individual's genome contains about 4 to 5 million SNPs. A major challenge in finding disease-related SNPs is identifying those that are synergistic or non-linearly related in causing a specific disease. While each SNP may have a relatively small or medium effect on a specific disease, the interaction of several SNPs can have a significant effect on that disease at the phenotype level. The study of the synergistic effect of SNPs on diseases is an important field of study in bioinformatics and computational biology. The main challenge in finding synergistic subsets of SNPs is a large number of possible solutions and a huge search space that is exponential concerning the number of SNPs. The problem of identifying SNP interactions can be defined as efficiently detecting multiple SNPs that are non-linearly related, significantly associated with phenotypes, and possibly influential in disease incidence. The search space for this problem contains all combinations of SNPs that may be related to a particular disease.

Despite using filters to limit the search space, the number of possible combinations for these SNPs is so large, and existing methods have only reached a few hundred or at most a few thousand SNPs combinations to check, and also only interactions of at most a limited k-order are searched. The calculation of interactions between these SNPs grows exponentially as the number of SNPs increases [1]. For example, to find the interactions of two SNPs (2-order), we need to examine all the binary combinations from the set including all the SNPs. Considering the higher orders, all of the combinations of 2, 3, . . ., k should be checked.

Given a dataset of m SNPs and n samples, each classified as a case (samples with the disease) or control (samples without the disease), the problem is to find one or more optimal sets of k SNPs (being k the epistasis order) that are influential on the disease state. For each SNP, there are three genotypes (possible values): the homozygous reference genotype, the heterozygous genotype, and the homozygous variant genotype. These genotypes are encoded into numeric labels (0, 1, and 2).

Given the genotype data of a set of samples, which are classified as cases and controls, the problem is to find a set of combinations of k SNPs (being k the epistasis size) that cause the disease. Since each SNP is represented with a number between 0 and $N-1$ (being N the number of SNPs in the samples dataset), a solution is given by a list of k non-repeated values between 0 and $N-1$.

In recent years, most of the related researches are limited to those that examine the interaction between only a few SNPs which still needs lots of computation and time, particularly when we deal with several hundred thousand SNPs. Machine learning [2] methods such as random forest [3–5], support vector machine (SVM) [6], association rules [7], and artificial neural networks [8, 9] have been used to extract epistasis interactions. From an ML perspective, finding the SNP-disease relationship is usually considered a classification problem, trying to find discriminatory features that classify the data properly.

Li et al. [10] proposed a permutation random forest (PRF) method, to detect SNP interactions. This method determines the effect of each pair of interacting SNPs on discriminating case and control samples by calculating $\Delta E = E_1 - E_2$, the difference between classification error when the original dataset is used ($E_1$), and the classification error for the permuted SNP pairs ($E_2$). Pairs with the highest $\Delta E$ are assumed to be interacting SNP pairs most related to the disease state.

Boutorh and Guessoum [11] proposed a method based on association rule mining to detect SNP interactions. Association rules are defined as rules in the form of A→B, that are frequent

(i.e. satisfy minimum support) in a transactional dataset. An association rule A→B implies that the occurrence of item A in a transaction is highly correlated with the occurrence of item B [12]. Since the traditional association rule mining methods such as Apriori, suffer from high computational costs, especially in datasets with a high number of features, the authors proposed a new algorithm based on Grammatical Evolution [13] to find SNP-SNP associations. Other machine learning approaches such as Bayesian neural network [14], multifactor dimensionality reduction (MDR) classification [15], and logistic regression [16] have also been considered for detecting disease-causing SNPs.

A wide variety of other methods such as clustering and information theory-based searching methods [17–21], statistical methods [2, 22–25] from which BOOST [26] is very well known, evolutionary and swarm intelligence-based algorithms such as MTACO-DMSI [27], and high-performance computing approaches [28–35] have also been used for the detection of SNP interactions in literature.

Since the searching of all possible SNPs combinations is not computationally efficient, different heuristic techniques such as MCMC-based search [22, 36], swarm intelligence search [37–40], harmony search-based algorithms [41–45], and evolutionary algorithms [46–48] have been proposed in the existing literature to speed up the search. Some of the existing methods have been suggested to limit the search space to low-order interactions. For instance, two harmony-search based methods, HS-MMGKG and FHSA-SED, are specifically designed to detect only 2-order interactions [41, 45]. Although heuristic search methods have been demonstrated to be relatively successful in detecting SNP epistasis interactions, they still show poor performance in diseases with no or low marginal effects. This means that the neighbor solutions (highly intersected subsets of SNPs) could not have contiguous fitness. In other words, SNP sets in the search space that includes both targeted and non-targeted-SNPs do not show a significant difference in fitness compared to SNP sets including only non-targeted-SNPs. For example, for a set of interacting SNPs containing 5 SNPs, the SNP combinations containing subsets of the target disease-causing 5-SNP set (a combination of 4 targeted SNPs and 1 non-targeted SNP), do not show a significant score compared to sets that do not contain any of the SNPs present in the target set. Consequently, the heuristic search algorithms may explore the neighborhood of this target SNP set, but due to a lack of improvement in the score it will ignore the neighborhood solutions and as a result, do not reach the true SNP combination.

In this study, we focus on addressing this problem by introducing an improved harmony search. In the improved harmony search, we take advantage of biological prior knowledge to improve the chance of biologically relevant SNP combinations to be generated in the initial harmonies. The proposed approach, namely locus-driven based harmony search (LDHS), efficiently generates solutions (harmonies) that are better candidates than a complete randomly generated set of individual solutions. Experimental results confirm that the proposed method improves the efficiency (in terms of time and number of detected SNP interactions) compared to the previous algorithms, including the HS algorithm without the proposed initialization.

In this study, we used genetic linkage, the pattern of simultaneous inheritance of parts of DNA, as biological evidence to guide the process of generating initial solutions. While the search algorithm may not be able to find the optimal SNP set in disease models with non-marginal effects, the proposed strategy for initializing harmony memories can instead compensate for this issue and generate better candidates that already guide the search algorithm to regions with higher scores. In the rest of this paper, first, the objective functions used in the previous research are introduced briefly in the methods section. After that, the biological background behind the new strategy used for generating individuals (besides random generation) is explained. We called this strategy SNP grouping which introduces groups of SNPs that biologically have a higher chance to interact and cause diseases. SNP grouping and weighting idea is

explained after the biological background is provided. Finally, the results of applying the proposed method on synthesized and real datasets are reported and discussed.

## 2 Materials and methods

In order to successfully detect high-order SNPs, it is essential to efficiently search the very large space of all possible SNP combinations. As mentioned in the introduction section, we propose a harmony search (HS) based method that integrates biological knowledge for generating harmonies with a higher chance of having interactions from a biological perspective. In this section, we first explain the objective functions and then explain the integration of biological knowledge.

In the present work, we used the same multi-population HS and objective functions as in Tuo et al. [42]. They proposed a multi-population HS-based method, namely MP-HS-DHSI, to detect high-order SNPs. Harmony search is a relatively new metaheuristic algorithm, proposed by Zong Woo Geem et al. [49], that is inspired by the process of selection of the ideal pitch by musicians, and has been successfully applied to solve different optimization problems [50, 51]. In HS, all feasible solutions (harmonies) are maintained in a memory called harmony memory (HM).

Different improvements on the original HS algorithm has been suggested by researchers, to improve HS performance; such as proposing new operators, parameter adaptation, improving HS by hybridizing it with other heuristic algorithms, integrating multiple objective functions and constraints into the original HS structure, and adaptation of new harmony memory initialization methods [52]. Multi-objective HS-based algorithms has been successfully proposed recently in order to address disease model-free SNP-interaction detection. It is shown in [42] that using four different objective functions (BN K2-score, LR-score, JS-score, and ND-JE score) enables the algorithm to discover SNP-disease interactions in different disease models. The idea of using different score functions is also used in other harmony-search based methods such as HS-MMGKG and HS-DP [43]. Although these four objective functions complement each other in being able to address different disease models, since the search space of SNP combinations is very large, according to reported results there are still conditions in which several disease-related SNPs are not detected. Visualizing landscapes related to the score of SNPs subsets reported in Fig 1 of [42] demonstrates an issue in current score functions; they do not address the disease with no marginal effects. One idea to address this issue in score functions is to improve or define score functions that discriminate between sets of SNPs containing a different number of correct disease-related SNPs, however, formulating a score function with this characteristic can be very complicated. In the present work, in addition to benefit from using a multi-objective HS, a new method for initialization of harmony memories is introduced. We used biological background knowledge to propose a new method for initialing the memory in Harmony search. The idea is that "SNPs with a higher probability of having interaction with each other from a biological perspective" should have a higher chance of being selected as a candidate in the algorithm to be included in disease-related SNP sets. As a result, the weakness of score functions in directing the search algorithm to successively add appropriate SNPs to the desired sets is compensated by creating individuals that already include SNPs with a high chance of having synergy among themselves. In the rest of this section, a brief overview of the score functions is provided, and then the biological background based on which the proposed Harmoney search with Locus-driven initialization (LDHS) method is introduced, and finally, the SNP grouping (and weighting) method which is used in harmony memory initialization is explained.

## 2.1 Objective functions

As mentioned, we used the same objective functions proposed by Tuo et al. [42]. These score functions are introduced briefly in the following subsections.

**2.1.1 BN K2-Score.** BN K2-score is a score used in Bayesian network models and is applied to evaluate the association between nodes that are connected using directed edges in an acyclic graph. This objective function has a high power to detect SNP interactions and is defined in Eq 1, as follows:

$$K2 - \text{Score} = \prod_{i=1}^{I} \frac{(j-1)!}{(n_i + j - 1)!} \prod_{j=1}^{J} n_{ij}! \tag{1}$$

in which j shows different states of the phenotype. In this study, because the data can be selected from two categories, j has two states, $n_i$ is the number of samples related to the genotype composition, and $n_{ij}$ is the number of samples obtained for phenotype j and genotype number i. The lower the value of the K2-Score is, the greater the association between an SNP combination and disease status.

**2.1.2 LR-Score.** LR-score is defined to measure some kind of relation between the disease and a specific SNP set [51, 53]. In order to estimate the strength of the relationship between each SNP to the disease state, this score uses the ratio between the observed and expected number of occurrences of each SNP in different disease states. If the observed value is equal to the expected value, we can conclude that the distribution of SNPs is completely random and independent of the disease state. The LR-Score function is defined as Eq (2).

$$LR = 2 \sum_{i=1}^{I} \sum_{j=1}^{J} o_{ij} \ln\left(\frac{o_{ij}}{e_{ij}}\right) = 2 \sum_{i=1}^{I} \sum_{j=1}^{J} n_{ij} \ln\left(\frac{n_{ij}}{e_{ij}}\right) \tag{2}$$

In this formula, $n_{ij}$ is the number of samples related to phenotype j and genotype i, and the $e_{ij}$ is the number of expected genotypes i having phenotype j. This function is well-compatible with the model of unknown diseases.

**2.1.3 JS-Score.** JS-score (Eq 3) is defined based on Kullback-Leibler (KL) divergence [54]. KL-divergence is an asymmetric distance measure defined to compare distributions. This objective function is used to examine the difference in the probability distribution between SNPs data with two different class labels.

$$JS = 0.5\left(\sum_{i=1}^{I} \sum_{j=1}^{2} \frac{n_{ij}}{n_i} \log\left(\frac{2n_{ij}}{n_i}\right)\right) \tag{3}$$

In this formula, $n_i$ is the number of samples related to genotype composition and $n_{ij}$ is the number of samples obtained for phenotype j, and genotype number i.

**2.1.4 Normalized distance with joint entropy.** The last objective function is the normalized distance with entropy (NDJE). This function, defined in Eqs 4–8) is proposed to enable the search algorithm to find combinations of SNPs that are associated with the disease state. SNP sets with low or no marginal effects can be better discovered by this score function. The idea behind this score function is that for a disease-causing SNP combination, there is a difference between the distribution of case and control samples, while for non-disease-causing SNP combinations, this difference is very small or zero.

$$NDJE = {}^{N}D(X) \big/ {}_{J}E(X_{\text{control}}) \tag{4}$$

$$ND(x) = \sum_{j=1}^{k} D(X_j) \Big/ D(X) \tag{5}$$

$$d(X_j) = \sum_{s=0}^{2} \sqrt{(n_s^{j.control} - n_s^{j.case})} 2 \qquad (6)$$

$$D(X) = \sum_{i=1}^{I} |n_i^{control} - n_i^{case}| \qquad (7)$$

$$JE(X_{control}) = -\sum_{i=1}^{I} p_i^{control} \log p_i^{control} = -\sum_{i=1}^{I} \frac{n_i^{control}}{n^{control}} \log \frac{n_i^{control}}{n^{control}} \qquad (8)$$

where X represents a k-order SNP combination including case and control samples, $X_{control}$ is a subset of X including only control samples; $n_s^{j.control}$ and $n_s^{j.case}$ indicate the number of samples with 'case' or 'control' state for $j^{th}$ SNP locus in which the SNP takes a value equal to s (homozygous major allele 0, heterozygous allele 1, and homozygous minor allele 2); $n_i^{control}$ and $n_i^{case}$ denotes the number of control and case samples in which the SNP combination X takes its $i^{th}$ value from all possible SNP combination values. Finally, the number of control samples is denoted by $n_i^{control}$. $d(x_j)$ (Eq 6) is the distance between distributions of case and control samples for $j^{th}$ SNP combination. D(X) is the difference between the numbers of case and control records for all SNP values in a SNP-combination X. JE ($X_{control}$) is used for normalization. The variable $p_i^{control}$ is the probability of observing a specific SNP combination with a value equal to i (the count of SNPs with the value i over the count of all SNP combinations).

## 3 Integration of biological knowledge

The problem with existing score functions is that they are not fully capable of modeling marginal effects, hence cannot discriminate between SNP sets containing no correct SNP and SNP sets containing only a subset of correct SNPs. The score function is only increased when the SNP combination contains all (and not only a subset) of the correct disease-causing SNPs. As a result, the search algorithm will not be guided to optimal points (since the neighboring regions of optimal points do not necessarily have near-optimal scores). To compensate for this lack of ability, we propose a SNP grouping prior to initial population generation using which SNPs with a higher probability of interaction are placed in each harmony (individual). The union of these SNPs naturally has a higher chance than random to be functional. This idea is explained in the next section.

### 3.1 SNP grouping

Genetic contents that are located close to each other on DNA sequence (on the same chromosome) have more tendency to be inherited together than independent genetic content. This phenomenon is called Genetic Linkage. Genetic Linkage has been so far known as a good source to detect genomic regions that affect diseases and as a result, genetic linkage studies have been so far used as one of the principal approaches to identify disease-causing genomic regions [55, 56]. While to the best of our knowledge, this biological phenomenon is ignored in the current methods available for identifying disease-causing SNPs, we use this idea to consider the SNP locus information in detecting these interacting SNPs. In addition, another intuition that makes the location of SNPs an important factor, is that SNP combinations that are disease-causing, should also be "inherited together", otherwise, if they are not inherited together, the combination would lose the chance to interact with each other, and as a result lose the chance to be able to cause the disease. A single gene and its promoter, which is located in the gene upstream, are close to each other on the DNA strand. So, they belong to a region with genetic linkage and are also functionally good candidates to have co-effects on diseases.

Three categories of SNPs can be considered: SNPs that occur in coding regions, SNPs that occur in non-coding regions, and those in the promoter region. SNPs can have different functions depending on where on the genome they are located. Mutations or SNPs that occur in the coding regions of genes cause changes in the proteins produced by genes, and groups of SNPs that occur in these regions, by possibly generating a synergic effect, can change the function of the protein and cause new traits (i.e. disease-causing functions) in the cell. Any significant error in the process of gene expression also leads to a change (such as a reduction or increment of amount, failure in production, etc.) in the final product. The gene expression process is initiated by the binding of transcription factors to the promoter region. Mutations in the promoter region can also lead to expression disorders.

The third category of SNPs, which are also more frequent on DNA than the first and second categories, are SNPs that occur in non-coding areas. Non-coding DNA has no instructions for making proteins and does not contain genetic information. Some non-coding regions of DNA, called introns, are located inside the genes but are removed, through RNA splicing, during the production of the final RNA strand.

We introduce a SNP grouping method based on the mentioned SNP categories in this section (i.e. the SNPs of the coding regions, the SNPs of the promoter regions, and the SNPs of the non-coding regions), and also considering the location of SNPs and the genetic linkage. This grouping of the SNPs based on their location, considering genetic linkage and functionality, provides sets of SNPs that are good candidates to be considered as disease-causing SNP combinations; and considering these groups, in addition to considering random SNP groups, can group together the SNPs that are not recognized in the search process when the non-marginal effect holds on the search space. The SNP information (location and category) is available in the SNP database on NCBI website (https://www.ncbi.nlm.nih.gov/snp/, Retrieved on November 3rd 2022).

Based on what was discussed in this section, in a preprocessing step, we use the NCBI SNP database to group all the SNPs. In other words, the input, which contains a set of SNPs, is searched in the NCBI SNP database. For each SNP, it is designated if it belongs to a gene-coding region, promotor region, or non-coding DNA. The SNPs are then grouped according to their natural location on DNA, SNPs that are close to each other (i.e. belong to a gene or its promoter) are put in the same group. These groups, then are used in a method explained further, in generating the initial harmonies.

## 3.2 Weighting SNPs

After retrieving the SNP information and grouping them, the number of vectors (harmonies or individual solutions) generated using each group is determined proportional to the number of functional or non-functional SNPs in each group. In the implementation, we considered two groups of SNPs. Group 1 includes SNPs that are located on genomic regions or gene upstream/downstream/promoter. Other SNPs are assigned to group 2. The number of generated harmonies from each group is proportional to the group size. In addition, we decreased the count of SNPs from group 2 by a factor of w, which we obtained using grid search and experiments on simulated data (w = 2). In the final population, the number of individuals that are generated using functional SNPs is more (experimentally specified) since functional SNPs have more chance to cause disease. Detail of the data preparations and memory initialization is represented in the pseudocode available in Fig 1.

As the outcome of this grouping and weighting scheme, the group of SNPs occurring on intron and intergenic regions take a smaller weight than genes on the promoter and also exon areas. Finally, in addition to random vectors, a set of vectors generated based on the proposed

```
input:      SNPs in the input dataset
             harmony_count = size of harmony memory to be generated
output:     A set of k-order SNPs combinations that are highly related to the disease state.

// step 1:  Identification of genes with SNPs located on their upstream/downstream/or coding regions
all_genes={}  // list of all SNP related genes
for each SNP in dataset
     gene = extract_from_NCBI(SNP)
     If gene != {} and all_genes not contain gene
         all_genes.add(gene, SNP)

// step 2: SNP Grouping
intron_ list = {}, Number_of_introns = 0
exon_ list = {}, Number_of_exons = 0
upstream_list = {}, Number_of_upstream = 0
total_number = 0
for each SNP
     if the location of the SNP was in an upstream
         upstream_ list.add(SNP)
         number_of_upstream = number_of_upstream +1

     if SNP is located on downstream and is not repeated
         downstream_list.add(SNP)
         number_of_downstream = number_of_downstream +1

     if SNP is located on exon and is not repeated
         exon_ list.add((SNP, gene))
         number_of_exons= number_of_exons +1

     if SNP is located on intron and is not repeated
         intron_list.add(SNP) //add tuple containing SNP and related gene
         number_of_introns= number_of_introns +1

         total_number= total_number+1
goup1_SNPs = upstream_ list + downstream_list + exon_ list
group2_SNPs = intron_list

//step 3: determining the proportion of each group of SNPs
//this weighting, helps the generated population to uniformly contain members from all genomic regions
Group1_size = (Number_of_upstream+ Number_of_downstream+ Number_of_exons) / Total_number
Group2_size = Number_of_introns / Total_number

//step 4: determine the size of each group in the initial population
w = experimentally extracted weight using grid search, for decreasing the proportion of group2_SNPs
//Initial population size of group 2
IPS_ group2 = (Group2_size / w) * harmony_count
//Initial population size of more important genomic regions
IPS_ group1 = (Group1_size + IPS_ group2) * harmony_count

// step 5 Construction of initial population vectors
for i in 1: IPS_ group1
     Generate a random harmony from group1_SNPs
     Adding to the initial memory if not repeated
for i in 1: IPS_ group2
     Generate a random harmony from group2_SNPs
     Adding to the initial memory if not repeated
return initial memory
```

**Fig 1.  The pseudocode related to the process of initial harmony generation, with considering SNP locus data.**
First, SNPs are grouped according to their genomic regions (gene upstream/downstream/coding or intron), and then number of harmonies belonging to each group is determined. Finally, the population is generated from SNPs.

grouping strategy are placed into the initial harmony memories. The flowchart of the proposed method is represented in Fig 2.

Co-occurrence of these SNPs that have biologically more chance to interact, increases the chance of a group of disease-causing SNPs to be placed in the same harmony in the harmony memory. This helps the search algorithm to initiate the HM with individuals that contain

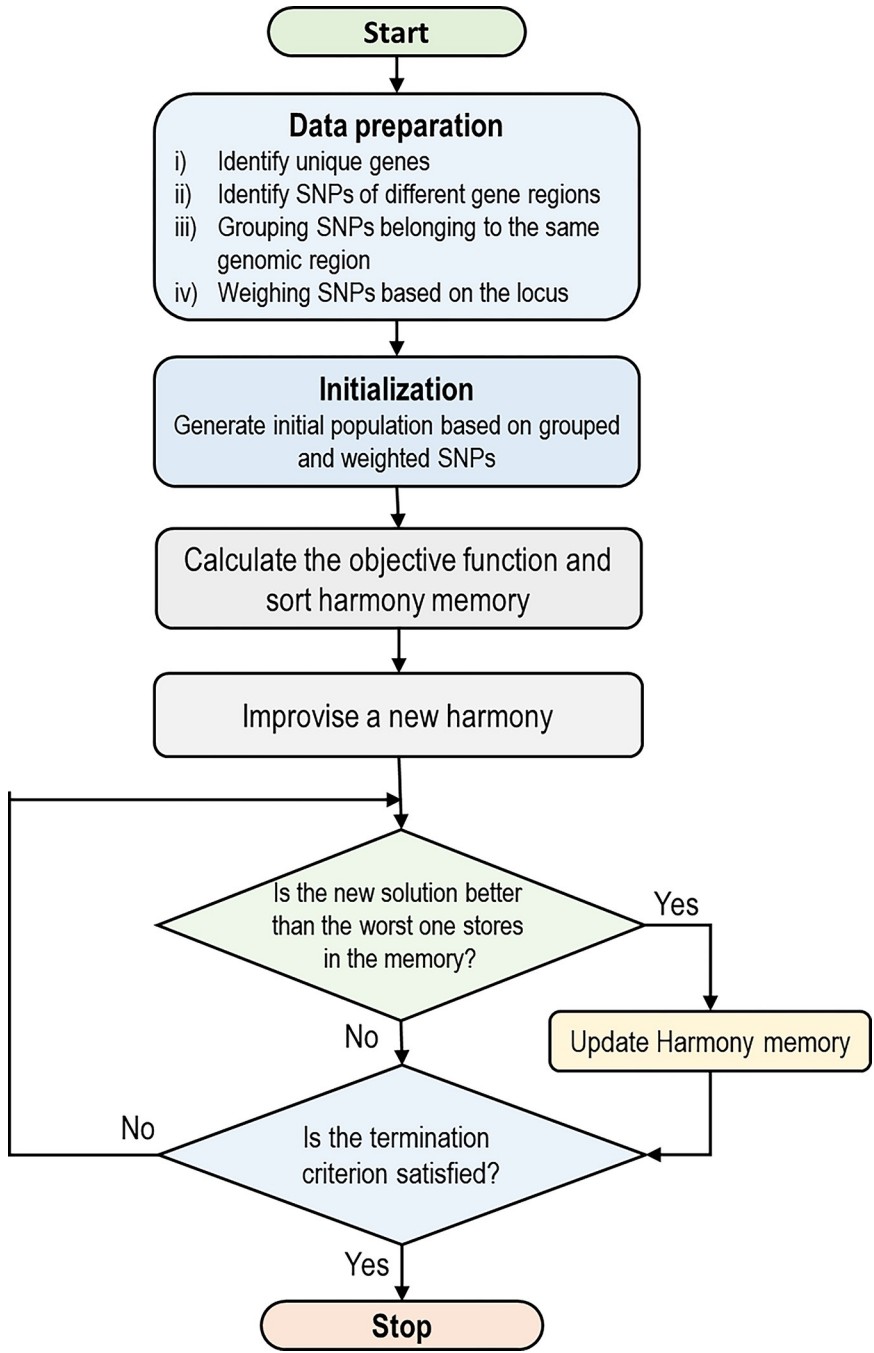

**Fig 2. Flowchart of steps in the proposed algorithm; specific data preparation and initialization based on SNP locus information and inheritance pattern of chromosomal regions are proposed to generate initial harmonies from SNP sets with a higher chance of interaction.**

SNPs having more chance to interact with each other from a biological perspective, which is beneficial for discovering SNPs for disease models with no marginal effects. Also, because of the same biological reason, it increases the chance to detect SNPs for disease models with marginal effects.

Grouping and weighting of 3rd, 4th and 5th-order SNP data were also performed to obtain multiple interactions. To do this, as in the case of grouping double SNPs, we first group the SNPs according to their natural position as explained formerly. Then the different SNPs of each gene are weighed and higher weight is considered for the SNPs that are in functional regions of the gene. After grouping, we generate the initial harmonies using these groups. It is possible to generate vectors of gene SNPs when the number of SNPs on the gene is greater than the number of interactions we have considered. When we have to choose an SNP but the number of SNPs in the neighborhood is not enough, we use the random selection of other SNPs. As it is observed in the experiments, the grouping of higher order SNPs also improves the searchability of the algorithm.

## 4 Experimental results

Synthesized and real datasets are used to evaluate the effectiveness of the proposed method. The results reported in this section, are achieved on MATLAB software, Windows X operating system with 4 GB of RAM, and a dual-core Intel processor.

We considered 8 other methods for comparison; the first one is a harmony search with similar objective functions but without the proposed initialization, MP-HS-DHSI [42]. Other algorithms, including CSE [37], MACOED [57], NHSA-DHSC [45], epiACO [58], and BEAM [59] are considered as swarm intelligence based highly cited and efficient methods that are highly used for comparisons in other similar work. MTHSA-DHEI [44] and MTACO-DMSI [27] are also considered two recent methods. The parameter settings for the algorithms used for comparison for all of the experiments are available in Table 1.

Table 1. Parameter setting for the algorithms used for comparison.

| Algorithm | Parameters | |
|---|---|---|
| BEAM (2007) | Default values | |
| CSE (2014) | The fraction of eggs discarded each generation | 0.25 |
| | The maximum number of steps to take in a levy flight | 1 |
| | The number of groups | 5 |
| | The number of nests (population size). | 100 |
| MACOED (2015) | antNumber (population size) | 50 |
| | $\rho$ | 0.9 |
| NHSA-DHSC (2017) | HMS (population size) | 50 |
| | HMCR | 0.95 |
| | PAR | 0.35 |
| epiACO (2017) | antNumber (population size) | 20 |
| | $\tau 0$ | 1 |
| | $\eta$ | 1 |
| | $\alpha$ | 1 |
| | $\beta$ | 1 |
| | evaporation coefficient $\rho$ | 0.2 |
| | constant $\xi$ | 0.3 |
| MP-HS-DHSI (2020) | HMS (population size) for each HM | 10 |
| | HMCR | 0.98 |
| | PAR | 0.35 |
| MTHSA-DHEI (2022) | HMCR | 0.98 |
| | HMS | 2*max(100, epi_dim*min(Dim/10,100)) |
| | PAR | 0.5 |
| MTACO-DMSI (2022) | HMS (population size) | 100 |
| | pre_error_rate | 0.45 |

## 4.1 Experiments on synthesized data

We considered two disease models for generating the synthesized dataset: with and without marginal effects. In the following subsections, for each of the two models, evaluation criteria including the number of successfully detected SNPs of order 2, 3, 4 and 5, the number of objective function evaluations, and execution time are provided.

**4.1.1 Disease models with marginal effects (DMEs).** Four disease models with marginal effects (DME 1–4) are considered for evaluation in this section. For each model, 100 simulated data sets with a sample size of 4000 (2000 controls and 2000 cases) were generated using GAMETES 2.1 software. The parameter settings are the same parameter setting and datasets used in MP-HS-DHSI [42].

Table 2 represents a comparison between the ability of the proposed algorithm and the mentioned 8 other methods considered for comparison. In order to keep the results consistent with reported results in Tuo, et. al, the results for CSE, NHSA-DHSC, EpiACO and BEAM algorithms are taken from the experimental results of MP-HS-DHSI paper result section. The other algorithms (publicly available code) were executed on the data, and their results are reported. Fig 3 represents the number of detected SNP interactions for each of the algorithms. As it can be observed, for SNP interactions of order 2 (DME-2), the method presented in this study (LDHS) is more powerful in detecting SNP interactions than the MPHS-DHSI method which uses the same objective functions but does not use the proposed initialization method. Also, on DME-3 and DME-4 it is comparable with the recent MTHSA_DHEI method (equal performance in DME-3 and 1% less in DME-4) and significantly performs better than all other algorithms, and on DME-1 and DME-2 it has a performance close to the best-performing method and also significantly better than the rest of the algorithms. Minimal dataset underlying diagram represented in Fig 3 is available in S1 Table.

Table 2 shows the average execution time of these methods. As it can be observed in Table 2, the initialization method has led to a significant improvement in the searchability of harmony search and resulted in a decrease in execution time. The new method converges fast compared to other algorithms and finds optimal solutions more quickly.

Finally, Table 3 shows the number of objective function evaluations performed in the proposed method (LDHS), exhaustive search, and the other swarm intelligence-based algorithms. The number of evaluations that must be done to complete the algorithm is minimum in the new method based on SNP grouping; i.e. SNP interactions are detected faster using the proposed method.

**4.1.2 Disease models with no marginal effects (DNMEs).** The best performance of the proposed method can be observed in diseases with no marginal effect. For each DNME, there are 100 datasets with 1500 case samples and 1500 controls. The data of nonfunctional SNPs are

**Table 2. Comparison of average execution time (seconds) of the new algorithm (LDHS), CSSE, NHSA-DHSC, epiACO, MP-HS-DHSI, and exhaustive search in detecting interactions of order 2.** Results of CSE, NHSA-DHSC, EpiACO, BEAM, and Exhaustive Search are taken from [42].

| Model | DME-1 | DME-2 | DME-3 | DME-4 |
|---|---|---|---|---|
| CSE | 1.21 | 1.22 | 1.18 | 1.15 |
| NHSA-DHSC | 0.12 | 0.13 | 0.44 | 0.97 |
| EpiACO | 0.57 | 0.66 | 0.69 | 0.74 |
| MP-HS-DHSI | 1.91 | 1.23 | 1.74 | 0.36 |
| BEAM | 1.14 | 1.14 | 1.14 | 1.14 |
| MTACO-DMSI | 814.3 | 532.5 | 5680 | 1304 |
| MTHSA_DHEI | 2.44 | 2.80 | 1.94 | 0.52 |
| Exhaustive Search | 2.13 | 2.01 | 2.12 | 2.05 |
| LDHS | 1.01 | 1.20 | 1.81 | 0.33 |

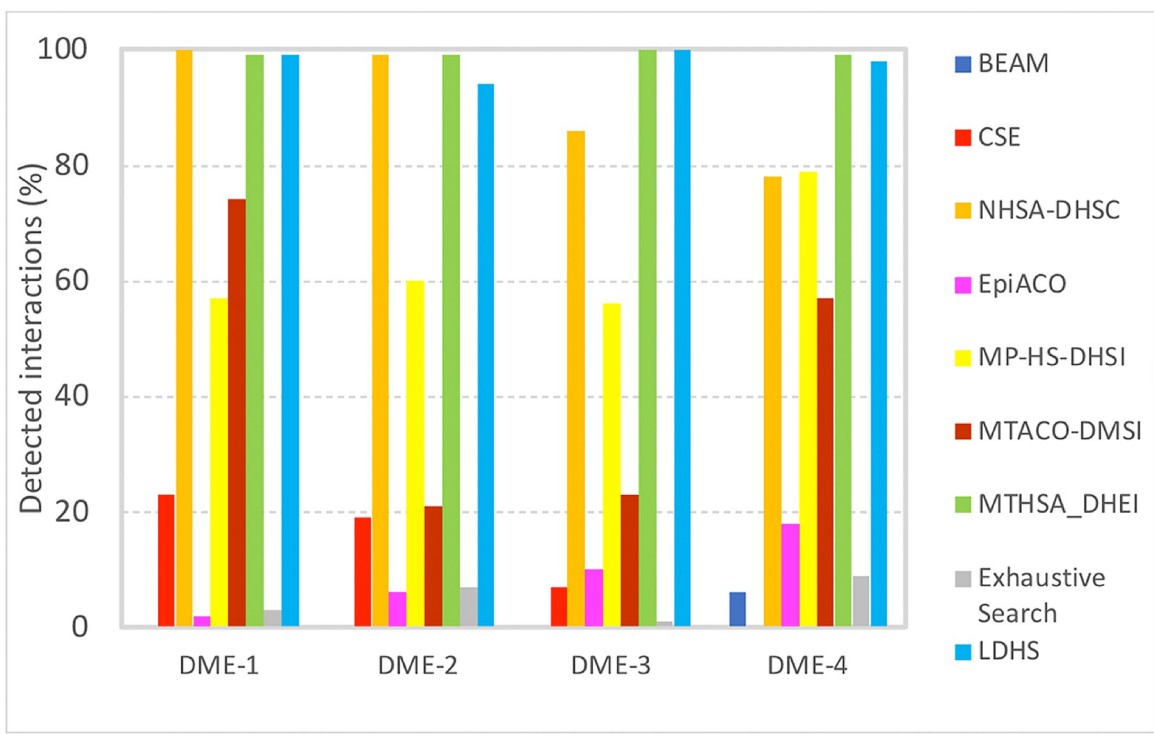

**Fig 3. Power of the proposed algorithm (LDHS) for detecting disease-causing interactions.** Results of CSE, NHSA-DHSC, EpiACO, BEAM and Exhaustive Search are taken from [42] in order to maintain consistency. For each model, 100 simulated datasets are used. For other algorithms the publicly available code is executed.

generated randomly based on the Hardy-Weinberg equilibrium (HWE). The parameter setting related to eight DNME models is similar to the parameter setting used in Tuo et al. [42], in order to maintain consistency in the results.

Fig 4 represents a comparison between the ability of the proposed method and other methods for finding SNPs of order 2 to 5 (represented by columns NM2 to NM5). The percentage of interactions that each algorithm is able to detect is reported. For detecting SNP interactions of order 2 to 5, the proposed method significantly outperforms the other methods, including the multi-population harmony search algorithm (with same objective functions) without the proposed initialization (i.e. MP-HS-DHSI). As it can be observed, even in case of 2-order interactions its performance is close to exhaustive search (96%). Minimal dataset underlying diagram represented in Fig 4 is available in S2 Table.

**Table 3. Comparison the number of objective function evaluations in proposed algorithm (LDHS), MP-HS-DHSI, exhaustive search, CSE, NHSA-DHSC, and epiACO in detecting 2-order interactions.** Results of CSE, NHSA-DHSC, EpiACO, BEAM and Exhaustive Search are taken from [42] in order to maintain consistency.

| Model | DME-1 | DME-2 | DME-3 | DME-4 |
|---|---|---|---|---|
| CSE | 2539 | 2540 | 2540 | 2540 |
| NHSA-DHSC | 240 | 242 | 751 | 1407 |
| EpiACO | 2024 | 2368 | 2479 | 2705 |
| MP-HS-DHSI | 2038 | 1337 | 1870 | 403 |
| MTACO-DMSI | 9380 | 5457 | 97000 | 65467 |
| MTHSA_DHEI | 455 | 1002 | 738 | 332 |
| Exhaustive Search | 4950 | 4950 | 4950 | 4950 |
| LDHS | 1078 | 1303 | 1943 | 365 |

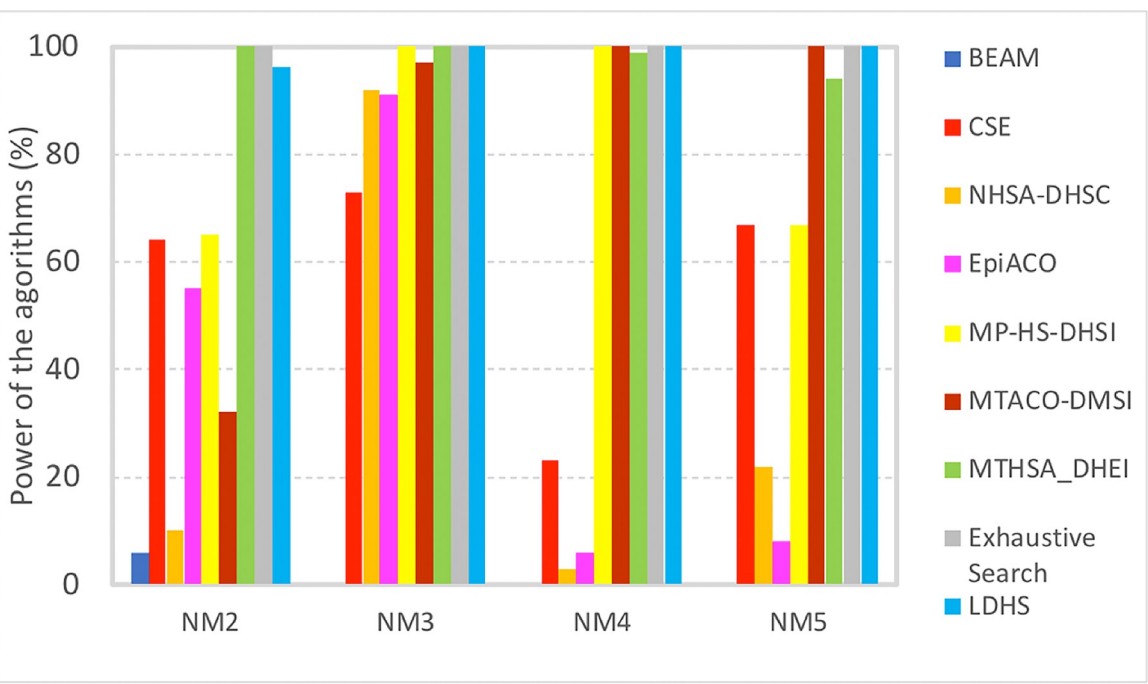

**Fig 4. Power of the proposed algorithm (LDHS) for detecting interactions of different orders.** Results of CSE, NHSA-DHSC, EpiACO, BEAM and Exhaustive Search are taken from [42] in order to maintain consistency. Other algorithms are executed with their default parameters (publicly available code).

In Table 4, the average execution time is reported for the proposed and other methods. Similar to Table 3, the running times of other methods are taken from [42]. As it can be observed in Table 4, the initialization method has led to a significant improvement in the searchability of the algorithm in the absence of the proposed initialization (i.e. MP-HS-DHSI) and resulted in a decrease in execution time. The new method converges faster than other algorithms and finds optimal solutions more quickly.

Finally, in Table 5 the number of objective function evaluations performed in the proposed method (LDHS), exhaustive search, and the other mentioned swarm intelligence-based algorithms is reported. The number of evaluations that must be done to complete the algorithm in LDHS is close to the best performing method and significantly lower than other methods; i.e. SNP interactions are detected faster using the proposed method.

## 4.2 Experiments on real data

The dataset of breast cancer [50, 60] used for evaluating previous works, containing 10,000 samples (5000 cases and 5000 controls), is considered for evaluating the effectivity of the

**Table 4. Comparison of average execution time (seconds) of the proposed algorithm (LDHS), CSE, NHSA-DHSC, epiACO, MTHSA-DHEI, MTACO-DMSI, MP-HS-DHSI and exhaustive search in detecting interactions of different orders.** Results of CSE, NHSA-DHSC, EpiACO, BEAM and Exhaustive Search are taken from [42]. Other algorithms are executed with their default parameters (publicly available code).

| Model | CSE | NHSA-DHSC | epiACO | MP-HS-DHSI | MTACO-DMSI | MTHSA_DHEI | Exhaustive search | LDHS |
|-------|-----|-----------|--------|------------|------------|------------|-------------------|------|
| NM2 | 137.8 | 218.2 | 47.4 | 63.4 | 640.3 | 15.2 | 202.7 | 33.2 |
| NM3 | 758.6 | 95.2 | 160.6 | 6.2 | 160.7 | 15.4 | 4916 | 2.9 |
| NM4 | 1053.2 | 601.7 | 162.3 | 8.9 | 55.4 | 9.6 | 4916 | 5.1 |
| NM5 | 735.9 | 591.2 | 286.4 | 340 | 137.3 | 15.01 | 4916 | 190 |

**Table 5. Comparison the number of objective function evaluations the proposed algorithm (LDHS), exhaustive search, CSE, NHSA-DHSC, and epiACO in detecting different order interactions.** Results of CSE, NHSA-DHSC, EpiACO, BEAM and Exhaustive Search are taken from [42] in order to maintain consistency.

| Model | CSE | NHSA-DHSC | epiACO | MP-HS-DHSI | MTACO-DMSI | MTHSA_DHEI | Exhaustive search | LDHS |
|-------|-----|-----------|--------|------------|------------|------------|-------------------|------|
| NM2 | 110743 | 81001 | 56960 | 27498 | 43647 | 450 | 161700 | 9400 |
| NM3 | 243166 | 34724 | 192102 | 2053 | 11898 | 1134 | 3921225 | 757 |
| NM4 | 339341 | 192001 | 187761 | 2877 | 2049 | 1863 | 3921225 | 1100 |
| NM5 | 356646 | 192001 | 364450 | 101338 | 4464 | 2495 | 3921225 | 40400 |

proposed method on detecting SNPs on real datasets. In this experiment, we compared the proposed algorithm with the harmony search algorithm with the same objective functions, MP-HS-DHSI, without the proposed initialization. Two SNP groups of order 2, i.e. (rs3020314, rs2017591) and (rs3020314, rs1514348) that was reported by Tuo et al., and satisfies significance levels (G-test and p-value of chi-square test), are also discovered also by our method after less number of objective function evaluations. The reported single SNPs with significant individual effects, SNPs of order 2 and 3 that were demonstrated to pass statistical tests, were also detected by the proposed method. As a result of applying the proposed initialization methods, the algorithm converged after a shorter number of objective function evaluations. Our experiments show that the average number of assessments needed to find the optimal SNP set is 409 without integrating biological prior knowledge. That's while the average number of assessments with the grouping-based generated harmonies added to the HMs beside random harmonies are only 71, which shows a considerable improvement. The search time is decreased from 1.29 to 0.22 seconds using the proposed strategy.

## 5 Discussion

In this work, we present an improved harmony search algorithm for detecting high-order interacting SNP combinations. LDHS is a method for improving metaheuristic search algorithms for detecting disease-causing SNPs. We demonstrate that in diseases with no marginal effect, where existing objective functions cannot effectively find optimal solutions, integrating biological knowledge can guide the search process to better candidates.

In the present study, our main focus is on addressing the issue of current score functions not being able to discriminate SNP sets that only contain a subset of one or more members (not all) of the true complete disease-causing SNPs. However, it is not straightforward to define or improve current objective functions to achieve this goal. As a result, we focus on initializing the harmony memories with SNP combinations that already have a higher chance of having interactions (in addition to random harmonies). This automatically increases the chance of putting together single SNPs that are parts of the final solution and therefore increases the chance of detecting disease-causing SNP combinations. The heuristic used in initializing the harmony memories is based on the biological fact that according to genetic linkage studies, SNPs located closer to each other on DNA strands have a greater tendency to have functional interactions and are a good source for detecting disease-causing genetic regions. In addition, we differentiate between SNPs located in coding or non-coding regions. Based on this knowledge, we propose a grouping strategy in advance of initializing harmony memories and propose a method based on this grouping to generate individual harmonies.

The idea behind the proposed method is that while the score functions are not efficient to discriminate the SNP sets with a different number of correct disease-related SNPs, initializing the correlating SNPs in one harmony in disease with non-marginal effects can increase the chance of the search algorithm to reach the optimum SNP set. Experimental results on simulated and synthesized datasets approve the effectiveness of the proposed method. 96% of two-

order SNP interactions and all five-order SNP interactions are detected by LDHS improved strategy, which shows a 31% and 33% improvement in the number of detected SNP combinations compared to MP-HS-DHSI algorithm which uses the same objective functions, but not benefit from biological knowledge integration in the harmony memory initialization. Three and four order SNP combinations are completely detected by the two algorithms. Also, the proposed method can perform close or equal to MTHSA-DHEI method which is a multitasking recent algorithm in disease models with no marginal effects, and better than MTHSA-D-HEI in disease models with marginal effects. LDHS outperforms all other compared algorithms (MP-HS-DHSI, CSE, MACOED, NHSA-DHSC, epiACO, BEAM, and MTA-CO-DMSI) in disease models with and without marginal effects.

As reported in the results section, the number of required objective function evaluations to find the correct answer is reduced compared to the harmony search with the same objective functions, and also in cases that harmony search without the proposed initialization has not been able to find some of the disease-causing SNP combinations, the harmony search with the proposed memory initialization has successfully detected the regarding SNP sets. Compared to other methods, the LDHS has a performance very close or better than the state-of-the art algorithms under study. Our experiments however, shows that the lowest number of objective function evaluations corresponds to MTHSA-DHEI which is a multitasking algorithm. LDHS is the second-best algorithm considering the number of objective function evaluations in most of the experiments. However, the idea in LDHS can also be applied using a multitasking strategy.

## 6 Conclusion

In this study, we consider the problem of discovering disease-causing SNP combinations. While there is a trend in using harmony search-based methods to detect high-order SNP combinations that cause a disease, studies show that these algorithms are not efficient in detecting SNP combinations with non-marginal effects. In diseases with non-marginal effects, common score functions cannot discriminate between SNP sets of size K that include different numbers of correct SNPs, and hence the search algorithm is not guided to optimal SNP sets. In this study, we propose a method for using biological knowledge to improve the memory initialization of harmony search. In addition to generating random harmonies, the proposed approach generates harmonies that include SNP combinations with a high tendency to interact considering the SNP locus information and the inheritance pattern of genomic regions. Experiments confirm that by adding these harmonies, the probability of finding optimal solutions in the presence of non-marginal effects increases. The experiments show that not only is the definition of objective functions important in finding disease-related SNPs, but also considering biological information can improve the performance of search algorithms. The authors suggest that this method can be applied to other metaheuristic algorithms for detecting SNP interactions.

## Supporting information

**S1 Table. Minimal dataset underlying Fig 2.**
(DOCX)

**S2 Table. Minimal dataset underlying Fig 3.**
(DOCX)

## Acknowledgments

We thank institute for advanced studies in basic sciences, for providing the facilities for doing this research.

## Author Contributions

**Conceptualization:** Fariba Esmaeili, Zahra Narimani, Mahdi Vasighi.

**Data curation:** Fariba Esmaeili.

**Investigation:** Fariba Esmaeili, Zahra Narimani.

**Methodology:** Fariba Esmaeili, Zahra Narimani, Mahdi Vasighi.

**Project administration:** Zahra Narimani.

**Software:** Fariba Esmaeili, Zahra Narimani, Mahdi Vasighi.

**Validation:** Fariba Esmaeili, Zahra Narimani, Mahdi Vasighi.

**Visualization:** Mahdi Vasighi.

**Writing – original draft:** Fariba Esmaeili, Zahra Narimani.

**Writing – review & editing:** Zahra Narimani, Mahdi Vasighi.

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
