## [Decision Letter · Decision Letter 0]

26 Jun 2023

PONE-D-23-13759Discovering SNP-disease relationships in genome-wide SNP data using an improved harmony search based on SNP locus and genetic inheritance patternsPLOS ONE

Dear Dr. Narimani,

Thank you for submitting your manuscript to PLOS ONE. After careful consideration, we feel that it has merit but does not fully meet PLOS ONE’s publication criteria as it currently stands. Therefore, we invite you to submit a revised version of the manuscript that addresses the points raised during the review process.

We look forward to receiving your revised manuscript.

Kind regards,

Sheng Du

Academic Editor

PLOS ONE

Journal Requirements:

3. Please upload a copy of Supporting Information Tables S2 and S3 which you refer to in your text on pages 18 and 20 (in PDF format).

**Additional Editor Comments:**

The paper has some application value. However, the author's contribution is not clearly formulated, and the algorithm is not expressed in detail. There are also many detail problems, and the authors are required to revise these carefully.

Reviewers' comments:

Reviewer's Responses to Questions

**Comments to the Author**

1. Is the manuscript technically sound, and do the data support the conclusions?

Reviewer #1: Yes

Reviewer #2: Partly

2. Has the statistical analysis been performed appropriately and rigorously? 

Reviewer #1: Yes

Reviewer #2: I Don't Know

3. Have the authors made all data underlying the findings in their manuscript fully available?

Reviewer #1: Yes

Reviewer #2: Yes

4. Is the manuscript presented in an intelligible fashion and written in standard English?

Reviewer #1: Yes

Reviewer #2: Yes

5. Review Comments to the Author

Reviewer #1: It appears that the paper successfully utilized harmony search in discovering SNP-disease relationships. However, it needs the following revisions:

= There are some more harmony search approaches applied in SNP. Please survey them also in literature section, such as:

- HS-MMGKG: A Fast Multi-objective Harmony Search Algorithm for Two-locus Model Detection in GWAS

- A Secure High-Order Gene Interaction Detecting Method for Infectious Diseases

= Section 2: Algorithm description is too short. Please provide algorithm pseudo-code and/or flow-chart. Also, provide the difference among algorithms in terms of algorithm structures by refering [1]:

[1] Review of Harmony Search with Respect to Algorithm Structure,” Swarm and Evolutionary Computation, 48, 31-43, 2019.

= Typos

- Table 1: In "HMCR = 0.98 0.98". Delete "= 0.98". Same for "Tau0=1, eta=1, alpha=1, beta=1"

Reviewer #2: The authors propose an algorithm (Discovering SNP-disease relationships in genome-wide SNP data using an improved harmony search based on SNP locus and genetic inheritance patterns) that claims to integrate biological knowledge to improve harmony memory, however, this is not reflected in the code, including the main contribution in the authors' paper (identifying SNPs of different gene regions, grouping SNPs. However, it is not reflected in the code, including the main contributions in the authors' paper (identifying SNPs of different gene regions, grouping SNPs, and weighting SNPs) are also not specific enough.

6. PLOS authors have the option to publish the peer review history of their article (what does this mean?). If published, this will include your full peer review and any attached files.

Reviewer #1: No

Reviewer #2: No

---

## [Author Response · Author response to Decision Letter 0]

27 Aug 2023

Dear Professor Sheng Du, 

Thanks for consideration and your comments on the submitted manuscript entitled “Discovering SNP-disease relationships in genome-wide SNP data using an improved harmony search based on SNP locus and genetic inheritance patterns”.

The formatting requirements addressed in PLOSOne_formatting_sample_main_body.pdf and PLOSOne_formatting_sample_title_authors_affiliations.pdf are applied to the re-submitted version. 

Minimal dataset is uploaded as a supplementary file (S1 and S2 Table).

Data (also code) is made available at public github repository, accessible via https://github.com/BioLab-IASBS/LDHS. Real data is also accessible via http://bioinfo.kmu.edu.tw/brca-steroid-96SNP.xlsx. Data availability statement is updated on re-submission. 

Supporting information with missing file is corrected (with adding reference to original data – explained in the manuscript added comments).

Authors contribution is explained in detail and the pseudocode is added (Fig 1).

The response to reviewer comments are provided in the following (also attached as a separate file as ‘Response to Reviewers’). For each comment, the answer is provided afterwards (beginning with a ‘++’ mark).

Sincerely,

Zahra Narimani

Department of Computer Science and Information Technology

Institute for Advanced Studies in Basic Sciences, Zanjan, Iran

Dear Reviewers, 

Authors would like to thank you for reviewing our manuscript entitled “Discovering SNP-disease relationships in genome-wide SNP data using an improved harmony search based on SNP locus and genetic inheritance patterns”.

The comments are addressed carefully and the answers are provided after each question (beginning with a ‘++’ mark). The most important comment was about missing detail of the algorithms. In addition to completing ‘section 3.2’ and adding the details of weighting and grouping method, the pseudocode of the proposed initialization is provided in Fig 1 of the revised manuscript. 

Sincerely,

F Esmaeili, Z Narimani, M Vasighi

Review Comments to the Author

Reviewer #1: 

It appears that the paper successfully utilized harmony search in discovering SNP-disease relationships. However, it needs the following revisions:

= There are some more harmony search approaches applied in SNP. Please survey them also in literature section, such as:

- HS-MMGKG: A Fast Multi-objective Harmony Search Algorithm for Two-locus Model Detection in GWAS

- A Secure High-Order Gene Interaction Detecting Method for Infectious Diseases

++ Thank you. The previous work section is updated in the manuscript (lines 86-90).

= Section 2: Algorithm description is too short. Please provide algorithm pseudo-code and/or flow-chart. 

++ Algorithm and flowchart are added (Fig 1 and Fig 2), also section “3.2 Weighting SNPs” is updated to add the details (lines 284-292). 

Also, provide the difference among algorithms in terms of algorithm structures by refering [1]:

[1] Review of Harmony Search with Respect to Algorithm Structure,” Swarm and Evolutionary Computation, 48, 31-43, 2019.

++ Thank you. This is a very interesting paper to use and address for explaining what is done in the proposed method. The explanation is added and highlighted in “2 Materials and methods”. Corrections can be found in lines 132-145. 

= Typos

- Table 1: In "HMCR = 0.98 0.98". Delete "= 0.98". Same for "Tau0=1, eta=1, alpha=1, beta=1"

++ corrected.

Reviewer #2:

The authors propose an algorithm (Discovering SNP-disease relationships in genome-wide SNP data using an improved harmony search based on SNP locus and genetic inheritance patterns) that claims to integrate biological knowledge to improve harmony memory, however, this is not reflected in the code, including the main contribution in the authors' paper (identifying SNPs of different gene regions, grouping SNPs. However, it is not reflected in the code, including the main contributions in the authors' paper (identifying SNPs of different gene regions, grouping SNPs, and weighting SNPs) are also not specific enough.

++ Thank you. The description of the process is updated (highlighted in the manuscript, section 3.2 - lines 284-292), and also the pseudocode of the proposed method is provided in Fig 1.

---

## [Decision Letter · Decision Letter 1]

18 Sep 2023

Discovering SNP-disease relationships in genome-wide SNP data using an improved harmony search based on SNP locus and genetic inheritance patterns

PONE-D-23-13759R1

Dear Dr. Narimani,

We’re pleased to inform you that your manuscript has been judged scientifically suitable for publication and will be formally accepted for publication once it meets all outstanding technical requirements.

Kind regards,

Sheng Du

Academic Editor

PLOS ONE

Additional Editor Comments (optional):

Reviewers' comments:

Reviewer's Responses to Questions

**Comments to the Author**

1. If the authors have adequately addressed your comments raised in a previous round of review and you feel that this manuscript is now acceptable for publication, you may indicate that here to bypass the “Comments to the Author” section, enter your conflict of interest statement in the “Confidential to Editor” section, and submit your "Accept" recommendation.

Reviewer #1: All comments have been addressed

Reviewer #2: All comments have been addressed

2. Is the manuscript technically sound, and do the data support the conclusions?

Reviewer #1: Yes

Reviewer #2: Yes

3. Has the statistical analysis been performed appropriately and rigorously? 

Reviewer #1: I Don't Know

Reviewer #2: I Don't Know

4. Have the authors made all data underlying the findings in their manuscript fully available?

Reviewer #1: Yes

Reviewer #2: No

5. Is the manuscript presented in an intelligible fashion and written in standard English?

Reviewer #1: Yes

Reviewer #2: Yes

6. Review Comments to the Author

Reviewer #1: The authors have addressed all the comments I raised as follows:

= more literature survey on harmony search approaches applied in SNP.

= strengthening algorithm description by providing algorithm pseudo-code and/or flow-chart.

= corrcting some typos.

Thus I recommend its acceptance.

Reviewer #2: The authors of the paper have made changes in accordance with the previous reviews, and I have only one small suggestion for the authors to give detailed document for using the software.

7. PLOS authors have the option to publish the peer review history of their article (what does this mean?). If published, this will include your full peer review and any attached files.

Reviewer #1: No

Reviewer #2: No

---

## [Editor Report · Acceptance letter]

6 Oct 2023

PONE-D-23-13759R1 

Discovering SNP-disease relationships in genome-wide SNP data using an improved harmony search based on SNP locus and genetic inheritance patterns 

Dear Dr. Narimani:

I'm pleased to inform you that your manuscript has been deemed suitable for publication in PLOS ONE. Congratulations! Your manuscript is now with our production department. 

Kind regards, 

on behalf of

Professor Sheng Du 

Academic Editor

PLOS ONE